# Preoperative Serum Markers and Risk Classification in Intrahepatic Cholangiocarcinoma: A Multicenter Retrospective Study

**DOI:** 10.3390/cancers14215459

**Published:** 2022-11-07

**Authors:** Masaki Kaibori, Kengo Yoshii, Hisashi Kosaka, Masato Ota, Koji Komeda, Masaki Ueno, Daisuke Hokutou, Hiroya Iida, Kosuke Matsui, Mitsugu Sekimoto

**Affiliations:** 1Department of Surgery, Kansai Medical University, Osaka 573-1191, Japan; 2Department of Mathematics and Statistics in Medical Sciences, Kyoto Prefectural University of Medicine, Kyoto 602-8566, Japan; 3Department of General and Gastroenterological Surgery, Osaka Medical College, Takatsuki 569-8686, Japan; 4Second Department of Surgery, Wakayama Medical University, Wakayama 641-8509, Japan; 5Department of Surgery, Nara Medical University, Kashihara 634-8521, Japan; 6Department of Surgery, Shiga University of Medical Science, Otsu 520-2192, Japan

**Keywords:** intrahepatic cholangiocarcinoma, preoperative serum marker, prognosis, classification and regression trees

## Abstract

**Simple Summary:**

We were able to stratify intrahepatic cholangiocellular carcinoma (ICC) patients who underwent hepatectomy into three risk groups using a classification and regression tree (CART) model for recurrence-free survival (RFS) and overall survival (OS). CART analysis, using results from the multivariate analysis, revealed decision trees for RFS and OS based on machine learning using preoperative serum markers. These three risk classifications using preoperative noninvasive prognostic factors could predict prognosis for ICC. These risk classifications are simple and easy to understand and can be clinically applied.

**Abstract:**

Accurate risk stratification selects patients who are expected to benefit most from surgery. This retrospective study enrolled 225 Japanese patients with intrahepatic cholangiocellular carcinoma (ICC) who underwent hepatectomy between January 2009 and December 2020 and identified preoperative blood test biomarkers to formulate a classification system that predicted prognosis. The optimal cut-off values of blood test parameters were determined by ROC curve analysis, with Cox univariate and multivariate analyses identifying prognostic factors. Risk classifications were established using classification and regression tree (CART) analysis. CART analysis revealed decision trees for recurrence-free survival (RFS) and overall survival (OS) and created three risk classifications based on machine learning of preoperative serum markers. Five-year rates differed significantly (*p* < 0.001) between groups: 60.4% (low-risk), 22.8% (moderate-risk), and 4.1% (high-risk) for RFS and 69.2% (low-risk), 32.3% (moderate-risk), and 9.2% (high-risk) for OS. No difference in OS was observed between patients in the low-risk group with or without postoperative adjuvant chemotherapy, although OS improved in the moderate group and was prolonged significantly in the high-risk group receiving chemotherapy. Stratification of patients with ICC who underwent hepatectomy into three risk groups for RFS and OS identified preoperative prognostic factors that predicted prognosis and were easy to understand and apply clinically.

## 1. Introduction

Intrahepatic cholangiocarcinoma (ICC) is the second most common primary liver cancer worldwide, with an increasing incidence over the last thirty years [1,2]. It is an aggressive neoplasm, and surgery is thought to be the only potentially curative treatment. However, the outcomes with this approach are usually poor, particularly in patients with regional lymph node metastases or positive surgical margins [3,4]. Certain elements of therapy, including resection range, lymphadenectomy extent, and adjuvant chemotherapy type, also remain controversial [5,6,7,8,9]. Therefore, risk stratification for selecting optimal candidates for surgery is crucial for patients with ICC. Because these patients typically have a poor prognosis, some physicians propose treatment with neoadjuvant chemotherapy or other nonsurgical treatment approaches [10]. In addition, prognostic tools have been developed to help identify patients at risk for a poor outcome after ICC resection and to inform preoperative decision-making regarding treatment [11,12,13]. However, most of these tools rely on pathological data and are not useful for application in preoperative settings [11,12,13]. Instead, useful biomarkers would be ones that are collected noninvasively and preoperatively. Many indices of nutrition, immunity, and inflammatory status have been found to be appropriate prognostic factors for various carcinomas [14,15,16,17,18,19]. These indices are based on preoperative findings and include the neutrophil-to-lymphocyte ratio (NLR) [20], prognostic nutritional index (PNI) [21], platelet-to-lymphocyte ratio (PLR) [22], C-reactive protein (CRP)-to-albumin ratio (CAR) [23], and CRP-albumin-lymphocyte (CALLY) index [24]. Given the likelihood of these inflammatory indicators to be prognostic for reduced overall survival (OS) and recurrence-free survival (RFS), pretreatment inflammatory indicators may serve as useful biomarkers for poor prognosis in patients with cholangiocarcinoma. These noninvasive biomarkers, however, remain insufficient for making decisions regarding treatment and life planning in patients with ICC.

Classification and regression tree (CART) analysis is a nonparametric decision tree technique that forms a collection of rules based on variables that can divide three populations into groups. CART analysis has gradually been incorporated into cancer prognosis prediction [25,26,27], and we have used it here to apply to a population of patients with ICC. Analyzing these indicators using CART and machine learning may improve the accuracy of biomarkers predicting ICC.

In this study, we aimed to establish biomarkers that accurately predict postoperative prognosis of patients with ICC from simple preoperative blood test data.

## 2. Materials and Methods

### 2.1. Patients

We retrospectively analyzed the clinical and histopathologic data of 225 consecutive patients with histologically confirmed ICC following hepatectomy at five university hospitals in the Kansai region of Japan between January 2009 and December 2020. Clinical data were collected from each hospital, then compiled at Kansai Medical University and analyzed at Kyoto Prefectural University of Medicine. This study was approved by the institutional review board of Kansai Medical University (approval number: 2019322). It was performed in accordance with the Declaration of Helsinki.

### 2.2. Clinicopathologic Variables

The patient factors investigated were age, gender, and presence of viral hepatitis. The blood test parameters examined were white blood cell (WBC), neutrophil, lymphocyte, hemoglobin, and platelet counts; albumin; total bilirubin; aspartate aminotransferase (AST); alanine aminotransferase (ALT); alkaline phosphatase (ALP) levels; CRP; and indocyanine green retention rate at 15 m (ICGR15). The following immunonutritive indices were also evaluated: NLR; PNI: (10 × albumin) + (0.005 × total lymphocyte); PLR; CAR; and integration of albumin-bilirubin (ALBI): (log10 [17.1 × bilirubin{mg/dL}] × 0.66) + (10 × albumin [g/dL] × [−0.085]). The surgical factors examined included surgical methods and operative blood loss, and the tumor factors examined were carcinoembryonic antigen (CEA), carbohydrate antigen (CA) 19-9, gross morphology of the tumor, tumor differentiation, metastasis to lymph nodes, tumor number, maximum tumor size, major vascular invasion, and postoperative adjuvant chemotherapy. At all institutions participating in this study, blood sampling for blood test parameters was performed from approximately 2 weeks to the day before surgery.

ICC staging was evaluated in accordance with the 8th edition of the AJCC staging system [28]. If there was distant metastasis, it was considered stage IV. Since we did not perform surgery on patients with distant metastases in our study, we included patients up to stage III. Regarding the type of hepatectomy, 0, 1, 2, and 3 indicated, respectively, less than mono-sectionectomy, mono-sectionectomy, bi-sectionectomy, and tri-sectionectomy.

### 2.3. Statistical Analysis

Continuous variables were divided into two groups according to cut-off values obtained by the receiver operating characteristic (ROC) curve analysis resulting in total mortality. Baseline patient characteristic comparisons of the three groups were performed using chi-square tests or Fisher’s exact tests, as appropriate. Risk classifications were established using CART analysis [29,30,31]. The CART algorithm split the data based on all available variables and classified based on the Gini Index and entropy criteria [32]. The CART algorithm was applied to classify patients into subgroups with similar prognoses using suitable factors classified based on cutoff values for ROC analysis. Using the ROC curve analysis and CART algorithm, patients were separated into three subgroups according to preoperative characteristics. The decision tree, a nonparametric supervised learning method, was applied to create a model that predicts the value of a target variable by learning simple decision rules and represents the results of the main operational relationships of each variable’s contribution to the outcome. A forced entry method was used to select preoperative marker factors for constructing the decision tree with recurrence as the outcome. For OS, the same items as for RFS were used for the first and second nodes. Risk classifications were made for RFS and OS rates after hepatectomy, and survival rates were calculated by the Kaplan–Meier method. The hazard ratio (HR) for RFS and OS and the 95% CIs were estimated using univariate and multivariate Cox hazard models. For all analyses, *p* values of less than 0.05 were used to denote statistical significance. All statistical analyses were performed with R version 4.1.2 (R Foundation for Statistical Computing, Vienna, Austria). Survival analysis was executed with R package “survival”. ROC curve analysis was executed with R package “pROC”. The risk classifications were established using CART analysis with the R package “rpart”.

## 3. Results

### 3.1. Preoperative Staging System using the CART Algorithm and Patients’ Perioperative Characteristics

Using preoperative prognostic factors and survival data, the CART algorithm divided patients into subgroups based on clinical prognosis. Twenty-three preoperative prognostic factors were extracted for the analysis, yielding a decision tree for recurrence with the top three factors (Figure 1). For RFS, CA19-9 was the first node, and CAR and CRP were the second nodes. The ROC curves and cutoff values for the three factors are shown in Appendix A. The numbers of patients in the low-, moderate-, and high-risk groups were 66, 98, and 61, respectively. The three risk classifications of patient characteristics are shown in Table 1. There were significant differences among the three groups in the following items: WBC, neutrophil, albumin, ALT, CRP, PNI, CAR, ALBI, CEA, CA19-9, type of hepatectomy, lymph node status, maximum tumor size, major vascular invasion, and AJCC staging.

Once the three risk classifications of survival were established, the Kaplan–Meier curves demonstrated that this classification enabled satisfactory risk evaluations of survival (Figure 2A,B). The RFS rates of the groups were significantly different for low-, moderate-, and high-risk groups, respectively, at 77.0%, 55.9%, and 36.3% for 1-year RFS; 66.1%, 29.2%, and 10.1% for 3-year RFS; and 60.4%, 22.8%, and 4.1% for 5-year RFS (moderate vs. low and high vs. low groups: *p* < 0.001) (Figure 2A). The patients were divided into three risk classifications of OS, and the OS rates of the groups significantly differed for low-, moderate-, and high-risk groups, respectively, at 96.9%, 87.3%, and 68.1% for 1-year OS; 80.9%, 52.3%, and 22.0% for 3-year OS; and 69.2%, 32.3%, and 9.2% for 5-year OS (moderate vs. low and high vs. low groups: *p* < 0.001) (Figure 2B).

As a comparison to our preoperative staging system, Kaplan–Meier curves for RFS and OS based on the 8th edition of the AJCC staging system are presented in Figure 2C,D. The 1-, 3-, and 5-year RFS rates of the groups were significantly different at 78.5%, 57.3%, and 41.9%; 59.7%, 31.4%, and 18.0%; and 54.5%, 22.4%, and 12.9% for stage I, II, and III, respectively (stage II vs. I and III vs. I groups: *p* < 0.001) (Figure 2C), and the 1-, 3-, and 5-year OS rates of the groups were significantly different at 95.0%, 90.1%, and 74.2%; 78.7%, 52.9%, and 32.5%; and 62.0%, 32.4%, and 19.5% for stage I, II, and III, respectively (stage II vs. I groups: *p* = 0.001 and III vs. I groups: *p* < 0.001) (Figure 2D). In comparing RFS, AIC was lower (1333 vs. 1356) and c-index higher (0.652 vs. 0.609) in our preoperative staging system versus the AJCC staging system. Again, AIC was lower (1033 vs. 1054) and C-index was higher (0.688 vs. 0.652) based on preoperative staging compared to AJCC staging.

### 3.2. Univariate and Multivariate Analysis of Prognostic Factors for Long-Term Survival

Cox proportional hazards analysis revealed seven independent prognostic predictors for both RFS and OS (Table 2): serum total bilirubin level > 0.67 mg/dL (RFS: HR, 1.62; 95% CI, 1.09–2.41; *p* = 0.018 and OS: HR, 1.81; 95% CI, 1.15–2.48; *p* = 0.013), serum ALP level > 298U/L (RFS: HR, 1.83; 95% CI, 1.20–2.79; *p* = 0.005 and OS: HR, 2.01; 95% CI, 1.24–3.26; *p* = 0.005), positive lymph node metastasis (RFS: HR, 1.97; 95% CI, 1.09–3.54; *p* = 0.024 and OS: HR, 3.48; 95% CI, 1.75–6.94; *p* < 0.001), tumor size ≥ 3.5 cm (RFS: HR, 1.91; 95% CI, 1.23–2.98; *p* = 0.004 and OS: HR, 2.55; 95% CI, 1.53–4.25; *p* < 0.001), AJCC stage II compared to I (RFS: HR, 1.89; 95% CI, 1.09–3.29; *p* = 0.024 and OS: HR, 2.04; 95% CI, 1.11–3.76; *p* = 0.022), moderate compared to low risk based on preoperative markers (RFS: HR, 2.93; 95% CI, 1.71–5.01; *p* < 0.001 and OS: HR, 3.75; 95% CI, 1.96–7.15; *p* < 0.001), and high compared to low risk based on preoperative markers (RFS: HR, 5.42; 95% CI, 2.88–10.22; *p* < 0.001 and OS: HR, 7.18; 95% CI, 3.33–15.46; *p* < 0.001).

### 3.3. Effects of Postoperative Adjuvant Chemotherapy in the Three Risk Groups

Figure 3 shows the results of RFS and OS based on postoperative adjuvant chemotherapy in the three risk groups. Although no differences were observed between patients with or without postoperative adjuvant chemotherapy in the low-risk group, postoperative adjuvant chemotherapy tended to improve OS in the moderate group (Figure 3D). Furthermore, postoperative adjuvant chemotherapy showed a significantly better effect of prolonging survival in the high-risk group (Figure 3F).

## 4. Discussion

Despite progress in the nonoperative management of ICC, its incidence has been increasing around the world [33]. Surgical resection is still the only strategy for potentially curative treatment [34,35]. However, the OS of patients selected to undergo this surgery generally remains poor, and almost two thirds of patients have a recurrence soon after surgery [36]. Several prognostic schemas and nomograms have been proposed to stratify the outcomes of patients with ICC [11,12,13], focused mainly on tumor-specific factors, including tumor number and size, major vascular invasion, lymph node status, and levels of serum CEA and CA19-9 [12,13]. Accurate prediction of outcomes and identification of patients who will benefit the most from surgery have become particularly important in patients with ICC. We employed a machine-based approach in the present study to identify subsets of patients with a distinct prognosis using preoperative data derived from a CART model. CART analysis using the results of multivariate analysis revealed that prognosis could be predicted based on values of CA19-9, CRP, and CAR as risk classifications for RFS. We also adapted the classifications established for predicting RFS to use for predicting OS. Our risk classification is characterized by its simplicity as it uses only the results of routine preoperative blood tests to predict prognosis.

Decision trees are widely used in healthcare and other professions for varied reasons, including easy interpretability. In contrast to other nontraceable neural network–based artificial intelligence models, which are increasingly employed in clinical practice, decision trees are comprehensible and reproducible for the user [37,38]. Therefore, the advantage of this risk classification based on CART analysis is that it is based on clinically sound, indispensable, and objective preoperative variables and is visually easy to understand. Furthermore, studies evaluating the prognostic value of CART analysis for predicting outcomes after hepatectomy using preoperative factors are limited. Performing this evaluation with preoperative factors was advantageous because decisions regarding surgical procedures can now be made based on these prognostic predictions. The AJCC staging system is classified using the TNM scoring system that includes tumor number and size, vascular invasion, visceral peritoneum invasion, and involvement of local extrahepatic tissue (T); number of involved regional lymph nodes (N); and presence or absence of distant metastases (M) [39]. The AJCC staging system is determined by pathological evaluation of resected tumors. In contrast, in this study we used only preoperative blood biochemical data for the three risk classifications to determine prognosis after surgery. The RFS and OS curves according to our three risk classification groups and the survival curves using the AJCC staging system were similar, but the AIC and C-index were slightly better for our three-risk classification system (Figure 1). CA19-9 level as a noninvasive biomarker is well known to reflect systemic tumor aggressiveness [40]. As for inflammation-nutrition-based markers, low levels of PNI [21], high levels of CAR [23], and high scores of GPS (score of 1/2) [41] and CONUT (score of more than 2) [42] have been associated with poor survival outcomes in resected ICC patients. In patients with malignancy, several studies exhibited prognostic utilities of various continuous inflammatory/immunonutritional markers including CAR, which aligns with our results [23,41,42]. 

The high-risk group classified by risk factors of CA19-9 and CRP had the worst prognosis for both RFS and OS. This high-risk group is characterized by the need for neoadjuvant and postoperative adjuvant chemotherapy. In this retrospective study, there were only a few patients who underwent neoadjuvant chemotherapy, and thus we did not evaluate the effect of neoadjuvant chemotherapy for ICC patients with high risk. The effect of postoperative adjuvant chemotherapy for these patients was investigated, and the survival rate was better in the treated group (Figure 3F). Prospective studies on whether to perform neoadjuvant chemotherapy and/or postoperative adjuvant chemotherapy for these high-risk patients is necessary in the future. We constructed a prognostic formula based on the Cox proportional hazards model, as follows: 1.33 × 10^−5^ CA19-9 + 2.03 CAR–0.48 CRP. In addition, they were classified into the following three groups based on the results of the prognostic prediction formula: −1.9 × 10^−1^ to 9.2 × 10^−4^, worse prognosis; 9.2 × 10^−4^ to 1.9 × 10^−2^, moderate prognosis; and 1.9 × 10^−2^ to 4.0, better prognosis. If the prognosis of a patient was predicted to worsen before surgery, it may be desirable to stop upfront surgery and perform chemotherapy. In addition, although it is necessary to consider in a prospective study, aggressive postoperative adjuvant chemotherapy may be warranted in the moderate prognosis group. However, this prognostic prediction formula and the results of three numerical classifications were predictions based on the results of our surgery, which was performed in only a small number of cases, and it is necessary to evaluate this formula by examining a large number of cases in the future.

The present study has some limitations. First, the study was subject to selection bias due to its retrospective nature. Second, although a strength of the study was the analysis of data from five universities in the Kansai region of Japan, this may have introduced some variation in patient selection and surgical techniques among the participating universities. Therefore, our results should be interpreted within the context of these limitations.

## 5. Conclusions

In conclusion, using a CART model, we were able to stratify patients with ICC who underwent hepatectomy into three risk groups for RFS and OS. These three risk classifications using preoperative noninvasive prognostic factors could predict prognosis. These risk classifications are simple and easy to understand and can be clinically applied.

## Figures and Tables

**Figure 1 cancers-14-05459-f001:**
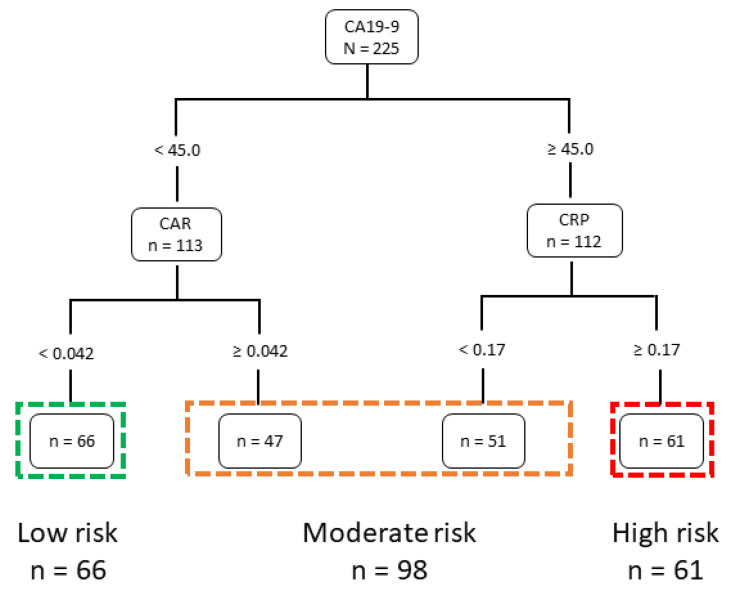
The decision tree model based on CA19-9, CAR, and CRP for predicting recurrence-free survival. CA19-9: carbohydrate antigen 19-9; CRP: C-reactive protein; CAR: CRP/albumin ratio.

**Figure 2 cancers-14-05459-f002:**
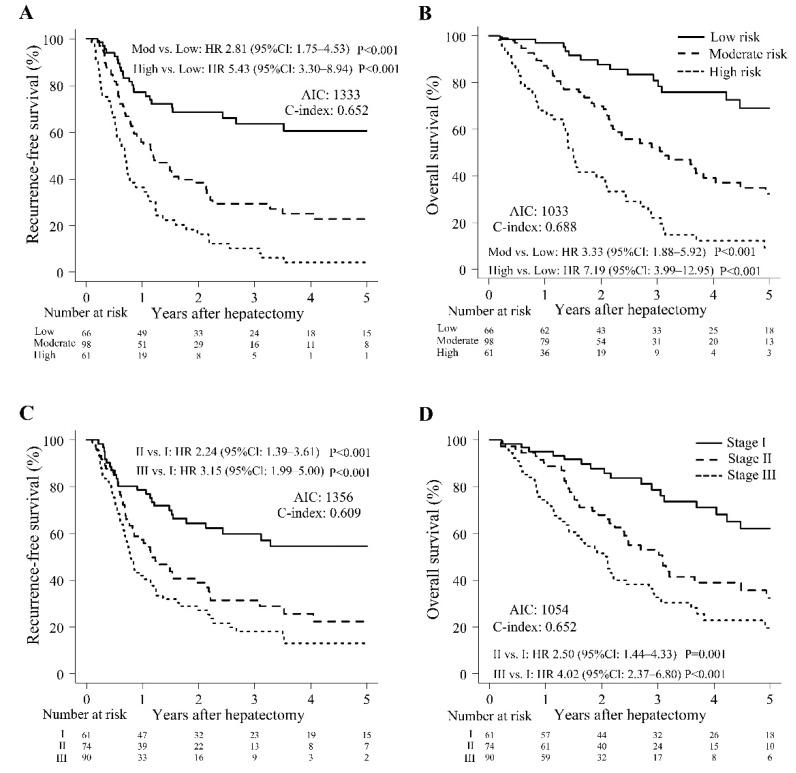
Survival outcomes. (**A**) Recurrence-free survival among three groups stratified according to the three risk classifications. (**B**) Overall survival among three groups stratified according to the three risk classifications. (**C**) Recurrence-free survival among three groups stratified according to the 8th edition of the AJCC staging system. (**D**) Overall survival among three groups stratified according to the 8th edition of the AJCC staging system. CI: confidence interval; HR: hepatic resection; AIC: Akaike information criterion.

**Figure 3 cancers-14-05459-f003:**
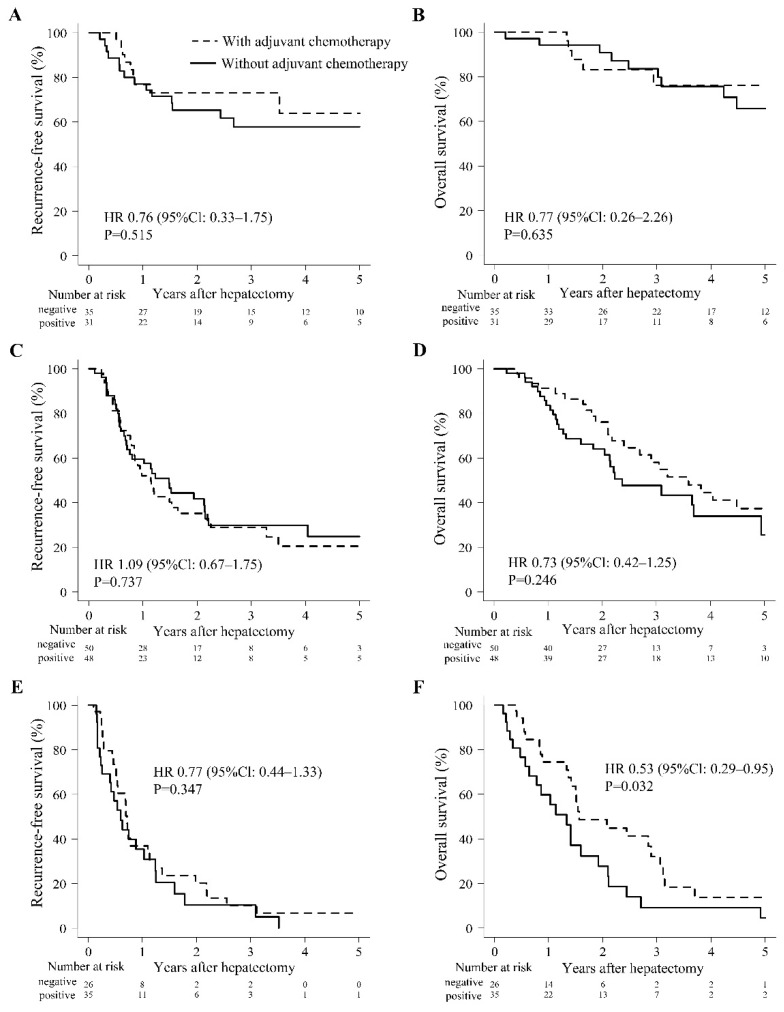
Comparison of survival outcomes after hepatic resection between patients with or without adjuvant chemotherapy. (**A**) Recurrence-free survival in the low-risk group. (**B**) Overall survival in the low-risk group. (**C**) Recurrence-free survival in the moderate-risk group. (**D**) Overall survival in the moderate-risk group. (**E**) Recurrence-free survival in the high-risk group. (**F**) Overall survival in the high-risk group. CI: confidence interval; HR: hepatic resection.

**Table 1 cancers-14-05459-t001:** Differences in clinical characteristics of 225 patients with intrahepatic cholangiocellular carcinoma classified into three risk groups by preoperative markers.

Variables	High Risk (*n* = 66)		Moderate Risk (*n* = 98)		Low Risk (*n* = 61)		*p*	Variables	High Risk (*n* = 66)		Moderate Risk (*n* = 98)		Low Risk (*n* = 61)		*p*
Age, years							0.257	PLR							0.666
<70	20	(30%)	42	(43%)	22	(36%)		<131.4	32	(48%)	46	(47%)	25	(41%)	
≥70	46	(70%)	56	(57%)	39	(64%)		≥131.4	34	(52%)	52	(53%)	36	(59%)	
Gender							0.091	CAR							**<0.001**
Male	50	(76%)	71	(72%)	36	(59%)		<0.042	66	(100%)	48	(49%)	3	(5%)	
Female	16	(24%)	27	(28%)	25	(41%)		≥0.042	0	(0%)	50	(51%)	58	(95%)	
HBsAg							0.507	ALBI							**<0.001**
Negative	55	(83%)	80	(82%)	54	(89%)		<2.82	44	(67%)	43	(44%)	18	(30%)	
Positive	11	(17%)	18	(18%)	7	(11%)		≥2.82	22	(33%)	55	(56%)	43	(70%)	
HCVAb							0.093	ICGR15 (%)							0.995
Negative	56	(85%)	91	(93%)	58	(95%)		<9.6	27	(43%)	41	(44%)	25	(43%)	
Positive	10	(15%)	7	(7%)	3	(5%)		≥9.6	36	(57%)	53	(56%)	33	(57%)	
WBC (/μL)							**<0.001**	CEA (ng/mL)							**0.006**
<5500	40	(61%)	48	(49%)	17	(28%)		<2.7	36	(55%)	48	(49%)	17	(28%)	
≥5500	26	(39%)	50	(51%)	44	(72%)		≥2.7	30	(45%)	50	(51%)	44	(72%)	
Neutrophil (/μL)							**0.013**	CA19-9 (U/mL)							**<0.001**
<3535	42	(64%)	48	(49%)	23	(38%)		<45	66	(100%)	47	(48%)	0	(0%)	
≥3535	24	(36%)	50	(51%)	38	(62%)		≥45	0	(0%)	51	(52%)	61	(100%)	
Lymphocyte (/μL)							0.837	Type of hepatectomy							**0.016**
<1347	29	(44%)	46	(47%)	30	(49%)		0 or 1	30	(45%)	35	(36%)	13	(21%)	
≥1347	37	(56%)	52	(53%)	31	(51%)		2 or 3	36	(55%)	63	(64%)	48	(79%)	
Hemoglobin (g/dL)							0.151	Operative blood loss (mL)							0.117
<12.8	26	(39%)	39	(40%)	33	(54%)		<540	36	(55%)	51	(52%)	23	(38%)	
≥12.8	40	(61%)	59	(60%)	28	(46%)		≥540	30	(45%)	47	(48%)	38	(62%)	
Platelet (×10^4^/μL)							0.210	Morphologic type							0.956
<20.1	38	(58%)	44	(45%)	27	(44%)		MF or IG	57	(86%)	82	(86%)	50	(85%)	
≥20.1	28	(42%)	54	(55%)	34	(56%)		PI or MF + PI or others	9	(14%)	13	(14%)	9	(15%)	
Albumin (g/dL)							**<0.001**	Differentiation							0.309
<4.0	15	(23%)	44	(45%)	33	(54%)		Well or moderate	55	(83%)	76	(78%)	53	(87%)	
≥4.0	51	(77%)	54	(55%)	28	(46%)		Poor or other	11	(17%)	22	(22%)	8	(13%)	
Total bilirubin (mg/dL)							0.460	Lymph node status							**<0.001**
<0.67	32	(48%)	44	(45%)	23	(38%)		Negative	64	(97%)	79	(81%)	36	(59%)	
≥0.67	34	(52%)	54	(55%)	38	(62%)		Positive	2	(3%)	19	(19%)	25	(41%)	
AST (IU/L)							0.326	Number of tumors							0.098
<25	29	(44%)	36	(37%)	19	(31%)		Solitary	60	(91%)	83	(85%)	47	(77%)	
≥25	37	(56%)	62	(63%)	42	(69%)		Multiple	6	(9%)	15	(15%)	14	(23%)	
ALT (IU/L)							**0.037**	Tumor size (cm)							**0.037**
<23	41	(62%)	43	(44%)	26	(43%)		<3.5	35	(53%)	46	(47%)	19	(31%)	
≥23	25	(38%)	55	(56%)	35	(57%)		≥3.5	31	(47%)	52	(53%)	42	(69%)	
ALP (U/L)							0.053	Major vascular invasion							**0.010**
<298	39	(59%)	49	(51%)	23	(38%)		Negative	63	(95%)	80	(82%)	47	(77%)	
≥298	27	(41%)	48	(49%)	38	(62%)		Positive	3	(5%)	18	(18%)	14	(23%)	
CRP (mg/dL)							**<0.001**	Adjuvant chemotherapy							0.456
<0.17	65	(98%)	52	(53%)	0	(0%)		No	35	(53%)	50	(51%)	26	(43%)	
≥0.17	1	(2%)	46	(47%)	61	(100%)		Yes	31	(47%)	48	(49%)	35	(57%)	
NLR							0.105	AJCC staging							**<0.001**
<2.4	35	(53%)	45	(46%)	21	(34%)		I	32	(48%)	24	(24%)	5	(8%)	
≥2.4	31	(47%)	53	(54%)	40	(66%)		II	22	(33%)	36	(37%)	16	(26%)	
PNI							**0.005**	III	12	(18%)	38	(39%)	40	(66%)	
<47	17	(26%)	41	(42%)	33	(54%)									
≥47	49	(74%)	57	(58%)	28	(46%)									

Data are shown as *n* (%). Bold indicates *p* < 0.05. HBsAg: hepatitis B surface antigen; HCVAb: hepatitis C virus antibody; WBC: white blood cell; AST: aspartate aminotransferase; ALT: alanine aminotransferase; ALP: alkaline phosphatase; CRP: C-reactive protein; NLR: neutrophil/lymphocyte ratio; PNI: prognostic nutritional index; PLR: platelet–lymphocyte ratio; CAR: CRP/albumin ratio; ALBI: integration of albumin-bilirubin; ICGR15: indocyanine green retention rate at 15 min; CEA: carcinoembryonic antigen; CA19-9: carbohydrate antigen 19-9; MF: mass forming; IG: intraductal growth; PI: periductal infiltrating; AJCC: American Joint Committee on Cancer.

**Table 2 cancers-14-05459-t002:** Cox proportional hazards regression analysis for recurrence-free survival and overall survival in patients with intrahepatic cholangiocellular carcinoma who underwent hepatic resection.

	Recurrence-Free Survival						Overall Survival					
Variables	Univariate Analysis			Multivariate Analysis			Univariate Analysis			Multivariate Analysis		
	HR	(95% CI)	*p*	HR	(95% CI)	*p*	HR	(95% CI)	*p*	HR	(95% CI)	*p*
Age ≥ 70 years (vs. <70 years)	0.76	(0.55–1.07)	0.113	0.98	(0.65–1.47)	0.910	0.81	(0.56–1.17)	0.263	1.08	(0.69–1.70)	0.724
Neutrophil ≥ 3535 (vs. <3535/μL)	1.16	(0.84–1.61)	0.372	0.93	(0.58–1.49)	0.776	1.07	(0.74–1.54)	0.731	0.72	(0.42–1.23)	0.235
Lymphocyte ≥ 1347 (vs. <1347/μL)	0.74	(0.53–1.03)	0.075	0.69	(0.41–1.14)	0.148	0.71	(0.49–1.02)	0.066	0.55	(0.30–1.00)	0.050
Platelet ≥ 20.1 (vs. <20.1 × 104/μL)	0.89	(0.64–1.24)	0.500	0.96	(0.58–1.59)	0.889	0.83	(0.57–1.20)	0.320	1.20	(0.67–2.14)	0.535
Albumin ≥ 4.0 (vs. <4.0 g/dL)	0.68	(0.49–0.95)	0.023				0.67	(0.46–0.97)	0.036			
Total bilirubin ≥ 0.67 (vs. <0.67 mg/dL)	1.55	(1.10–2.18)	0.011	1.62	(1.09–2.41)	0.018	1.69	(1.15–2.48)	0.008	1.81	(1.13–2.88)	0.013
ALT ≥ 23 (vs. <23 IU/L)	1.07	(0.77–1.48)	0.691	0.75	(0.48–1.17)	0.209	1.33	(0.92–1.92)	0.133	0.76	(0.47–1.25)	0.282
ALP ≥ 298 (vs. <298 U/L)	1.48	(1.06–2.06)	0.021	1.83	(1.20–2.79)	0.005	1.44	(1.00–2.09)	0.052	2.01	(1.24–3.26)	0.005
CRP ≥ 0.17 (vs. <0.17)	2.55	(1.82–3.57)	<0.001				2.49	(1.71–3.63)	< 0.001			
NLR ≥ 2.4 (vs. <2.4)	1.33	(0.95–1.85)	0.093	1.03	(0.60–1.76)	0.927	1.18	(0.82–1.71)	0.375	0.96	(0.52–1.80)	0.909
PNI ≥ 47.0 (vs. <47.0)	0.63	(0.45–0.88)	0.007	0.75	(0.42–1.36)	0.349	0.60	(0.42–0.88)	0.008	0.65	(0.32–1.32)	0.235
PLR ≥ 131.4 (vs. <131.4)	0.87	(0.63–1.21)	0.400	0.64	(0.38–1.10)	0.105	0.81	(0.56–1.17)	0.263	0.54	(0.29–1.01)	0.056
CAR ≥ 0.042 (vs. <0.042)	2.79	(1.99–3.91)	<0.001				2.67	(1.83–3.89)	<0.001			
ALBI ≥ 2.82 (vs. <2.82)	1.71	(1.23–2.39)	0.002	0.93	(0.53–1.62)	0.786	1.72	(1.18–2.50)	0.005	0.67	(0.35–1.28)	0.222
ICGR15 ≥ 9.6 (vs. <9.6 %)	0.96	(0.69–1.35)	0.833	0.88	(0.58–1.34)	0.566	1.11	(0.76–1.61)	0.601	0.95	(0.59–1.51)	0.815
CEA ≥ 2.7 (vs. <2.7 ng/mL)	1.52	(1.09–2.14)	0.015	0.86	(0.57–1.31)	0.492	1.59	(1.09–2.33)	0.017	1.03	(0.64–1.67)	0.900
CA19-9 ≥ 45.0 (vs. <45.0 U/mL)	2.19	(1.56–3.07)	<0.001				2.93	(1.98–4.33)	<0.001			
Type of hepatectomy 2 or 3 (vs. 0 or 1)	1.08	(0.76–1.52)	0.677	0.66	(0.41–1.05)	0.082	1.07	(0.73–1.57)	0.726	0.47	(0.27–0.81)	0.006
Blood loss ≥ 540 (vs. <540 mL)	1.78	(1.27–2.49)	<0.001	1.29	(0.84–1.99)	0.240	1.68	(1.15–2.45)	0.007	1.10	(0.67–1.80)	0.699
Morphologic type PI or MF + PI or others (vs. MF or IG)	0.75	(0.44–1.26)	0.272	0.77	(0.41–1.44)	0.414	0.95	(0.55–1.64)	0.849	1.26	(0.62–2.55)	0.524
Differentiation, poor or others (vs. well or moderate)	1.56	(1.04–2.36)	0.033	0.99	(0.60–1.65)	0.978	1.18	(0.73–1.90)	0.493	0.66	(0.36–1.20)	0.173
Lymph node status, positive (vs. negative)	2.98	(2.06–4.31)	<0.001	1.97	(1.09–3.54)	0.024	3.63	(2.43–5.41)	<0.001	3.48	(1.75–6.94)	<0.001
Number of tumors, multiple (vs. solitary)	2.47	(1.65–3.69)	<0.001	1.49	(0.90–2.46)	0.120	2.33	(1.51–3.58)	<0.001	1.83	(1.08–3.09)	0.024
Tumor size ≥ 3.5 (vs. <3.5 cm)	2.07	(1.47–2.92)	<0.001	1.91	(1.23–2.98)	0.004	2.11	(1.44–3.10)	<0.001	2.55	(1.53–4.25)	<0.001
Major vascular invasion, positive (vs. negative)	2.45	(1.63–3.69)	<0.001	1.34	(0.79–2.29)	0.279	2.36	(1.48–3.75)	<0.001	1.51	(0.81–2.79)	0.191
Adjuvant chemotherapy, yes (vs. no)	1.04	(0.75–1.45)	0.802	0.89	(0.60–1.33)	0.565	0.85	(0.59–1.23)	0.399	0.59	(0.37–0.92)	0.020
AJCC staging II (vs. I)	2.24	(1.39–3.61)	<0.001	1.89	(1.09–3.29)	0.024	2.50	(1.44–4.33)	0.001	2.04	(1.11–3.76)	0.022
AJCC staging III (vs. I)	3.15	(1.99–5.00)	<0.001	1.16	(0.60–2.22)	0.658	4.02	(2.37–6.80)	<0.001	1.21	(0.55–2.66)	0.632
Preoperative marker, moderate (vs. low)	2.81	(1.75–4.53)	<0.001	2.93	(1.71–5.01)	<0.001	3.33	(1.88–5.92)	<0.001	3.75	(1.96–7.15)	<0.001
Preoperative marker, high (vs. low)	5.43	(3.30–8.94)	<0.001	5.42	(2.88–10.22)	<0.001	7.19	(3.99–12.95)	<0.001	7.18	(3.33–15.46)	<0.001

HR: hazard ratio; ALT: alanine aminotransferase; ALP: alkaline phosphatase; CRP: C-reactive protein; NLR: neutrophil/lymphocyte ratio; PNI: prognostic nutritional index; PLR: Platelet-lymphocyte ratio; CAR: CRP/albumin ratio; ALBI: integration of albumin-bilirubin; ICGR15: indocyanine green retention rate at 15 min; CEA: carcinoembryonic antigen; CA19-9: carbohydrate antigen 19-9; MF: mass forming; IG: intraductal growth; PI: periductal infiltrating; AJCC: American Joint Committee on Cancer.

## Data Availability

Due to the nature of this research, participants in this study could not be contacted regarding whether the findings could be shared publicly, thus supporting data are not available. The datasets generated and/or analyzed for the current study are not publicly available due to the nature of the research, as noted above.

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
