# Peer review of "Preoperative Serum Markers and Risk Classification in Intrahepatic Cholangiocarcinoma: A Multicenter Retrospective Study"

_cancers, 2022, doi:10.3390/cancers14215459_

Round 1

Reviewer 1 Report

The manuscript revealed decision trees for RFS and OS using CART analysis and created three risk classifications based on machine learning of preoperative serum markers to stratify ICC patients into three group preoperatively, which may help to selects patients who benefit most from surgery. This study rely on blood test parameters in preoperative settings, and the follow-up period is lengthy and convincing. However, some suggestions and questions are as follows:

1.      Some of the contents of the introduction, such as relevant inflammatory indicators, can be briefly described.

2.      The goal of this study is to use the CART model to stratify the risk of ICC patients and to investigate the impact of various serum markers on prognosis. But I didn't see how this study's CART model was constructed? Why are CA199, CAR, and CRP critical nodes? And how do you find out what their cut-off values are? Please provide some explanations or cite relevant sources.

3.      The findings revealed significant differences in RFS and OS rates between high, medium, and low risk groups, as well as AJCC stages III, II, and I, respectively. So, what are the distinctions between the high and medium risk groups, as well as the stage III and stage II groups?

4.      The goal of this study is to identify patients who will benefit the most from surgery, please include some discussion about the surgical strategy or clinical significance. This will increase the clinical value and applicability of your research.

5.      Moreover, minor editing and language revisions are needed.

If the author can reply to the above questions and make modifications, my opinion is to make minor revisions. Please revise the manuscript according to the above comments.

Author Response

We thank the reviewer for the valuable comments.

Comment

  1. Some of the contents of the introduction, such as relevant inflammatory indicators, can be briefly described.

Response

We appreciate this comment from the reviewer. Accordingly, we have inserted a reference for each indicator and added an explanation as follows:

Introduction (lines 61–68)

These indices are based on preoperative findings and include the neutrophil-to-lymphocyte ratio (NLR) [20], prognostic nutritional index (PNI) [21], platelet-to-lymphocyte ratio (PLR) [22], C-reactive protein (CRP)-to-albumin ratio (CAR) [23], and CRP-albumin-lymphocyte (CALLY) index [24]. Given the likelihood of these inflammatory indicators to be prognostic for reduced overall survival (OS) and recurrence-free survival (RFS), pretreatment inflammatory indicators may serve as useful biomarkers for poor prognosis in patients with cholangiocarcinoma.

  1. The goal of this study is to use the CART model to stratify the risk of ICC patients and to investigate the impact of various serum markers on prognosis. But I didn't see how this study's CART model was constructed? Why are CA199, CAR, and CRP critical nodes? And how do you find out what their cut-off values are? Please provide some explanations or cite relevant sources.

Response

We thank the reviewer for this comment. The CART algorithm split the data based on all available variables and classified based on the Gini Index and entropy criteria [Grajski 1986; new ref. no. 32]. The CART algorithm was applied to classify patients into subgroups with similar prognoses using suitable factors classified based on cutoff values from ROC analysis. Using the ROC curve analysis and CART algorithm, patients were separated into three subgroups according to preoperative characteristics. Twenty-three preoperative prognostic factors were extracted for the analysis, yielding the decision tree for recurrence with the top three factors. The ROC curves and cutoff values for the three factors are shown in Supplementary Figure 1.

Methods (lines108–131)

2.3. Statistical analysis

Continuous variables were divided into two groups according to cut-off values obtained by the receiver operating characteristic (ROC) curve analysis resulting in total mortality. Baseline patient characteristic comparisons of the three groups were performed using chi-square tests or Fisher's exact tests, as appropriate. Risk classifications were established using CART analysis [29-31]. The CART algorithm split the data based on all available variables and classified based on the Gini Index and entropy criteria [32]. The CART algorithm was applied to classify patients into subgroups with similar prognoses using suitable factors classified based on cutoff values for ROC analysis. Using the ROC curve analysis and CART algorithm, patients were separated into three subgroups according to preoperative characteristics. The decision tree, a nonparametric supervised learning method, was applied to create a model that predicts the value of a target variable by learning simple decision rules and represents the results of the main operational relationships of each variable’s contribution to the outcome. A forced entry method was used to select preoperative marker factors for constructing the decision tree with recurrence as the outcome. For OS, the same items as for RFS were used for the first and second nodes. Risk classifications were made for RFS and OS rates after hepatectomy, and survival rates were calculated by the Kaplan-Meier method. The hazard ratio (HR) for RFS and OS and the 95% CIs were estimated using univariate and multivariate Cox hazard models. For all analyses, P values of less than 0.05 were used to denote statistical significance. All statistical analyses were performed with R version 4.1.2 (R Foundation for Statistical Computing, Vienna, Austria). Survival analysis was executed with R package “survival.” ROC curve analysis was executed with R package “pROC.” The risk classifications were established using CART analysis with the R package “rpart.”

New references in the revised manuscript:

  1. Cao, F.; Shen, L.; Qi, H.; Xie, L.; Song, Z.; Chen, S.; Fan, W. Tree-based classification system incorporating the HVTT-PVTT score for personalized management of hepatocellular carcinoma patients with macroscopic vascular invasion. Aging 2019, 11, 9544–9555.
  2. Shimose, S.; Tanaka, M.; Iwamoto, H.; Niizeki, T.; Shirono, T.; Aino, H.; Noda, Y.; Kamachi, N.; Okamura, S.; Nakano, M.; et al. Prognostic impact of transcatheter arterial chemoembolization (TACE) combined with radiofrequency ablation in patients with unresectable hepatocellular carcinoma: comparison with TACE alone using decision-tree analysis after propensity score matching. Hepatol. Res. 2019, 49, 919–928.
  3. Feng, L.H.; Sun, H.C.; Zhu, X.D.; Liu, X.F.; Zhang, S.Z.; Li, X.L.; Li, Y.; Tang, Z.Y. Prognostic nomograms and risk classifications of outcomes in very early-stage hepatocellular carcinoma patients after hepatectomy. Eur. J. Surg. Oncol. 2021, 47, 681–689.
  4. Grajski, K.A.; Breiman, L.; Viana Di Prisco, G.; Freeman, W.J.; et al.: Classification of EEG spatial patterns with a tree-structured methodology: CART. IEEE Trans. Biomed. Eng. 1986, 33, 1076–1086.

Results (lines132–145)

  1. Results

3.1. Preoperative staging system using the CART algorithm and patients’ perioperative characteristics

Using preoperative prognostic factors and survival data, the CART algorithm divided patients into subgroups based on clinical prognosis. Twenty-three preoperative prognostic factors were extracted for the analysis, yielding a decision tree for recurrence with the top three factors (Figure 1). For RFS, CA19-9 was the first node, and CAR and CRP were the second nodes. The ROC curves and cutoff values for the three factors are shown in Supplementary Figure 1. The numbers of patients in the low-, moderate-, high-risk groups were 66, 98, and 61, respectively. The three risk classifications of patient characteristics are shown in Table 1. There were significant differences among the three groups in the following items: WBC, neutrophil, albumin, ALT, CRP, PNI, CAR, ALBI, CEA, CA19-9, type of hepatectomy, lymph node status, maximum tumor size, major vascular invasion, and AJCC staging.

Supplementary Figure 1

  1. The findings revealed significant differences in RFS and OS rates between high, medium, and low risk groups, as well as AJCC stages III, II, and I, respectively. So, what are the distinctions between the high and medium risk groups, as well as the stage III and stage II groups?

Response

We appreciate this comment from the reviewer. We used univariate and multivariate Cox hazard models with dummy variables. Therefore, it was impossible to use the Cox hazard model for explanatory variables containing three categories to avoid multicollinearity. Accordingly, we reduced the number of variables by one and performed the analysis.

For reference, the results of the multivariate Cox hazard model based on the AJCC staging â…¢ group or high-risk group are shown below.

Recurrence-free survival, HR (95% CI), P value

 AJCC staging â…  (versus â…¢): 0.86 (0.45–1.66), P = 0.658

 AJCC staging â…¡ (versus â…¢): 1.64 (0.96–2.78), P = 0.069

 Preoperative marker, low (versus high): 0.18 (0.10–0.35), P < 0.001

 Preoperative marker, moderate (versus high): 0.54 (0.34–0.85), P = 0.008

Overall survival, HR (95% CI), P value

 AJCC staging â…  (versus â…¢): 0.82 (0.38–1.81), P = 0.632

 AJCC staging â…¡ (versus â…¢): 1.69 (0.87–3.25), P = 0.119

 Preoperative marker, low (versus high): 0.14 (0.06–0.30), P < 0.001

 Preoperative marker, moderate (versus high): 0.52 (0.30–0.90), P = 0.020

  1. The goal of this study is to identify patients who will benefit the most from surgery, please include some discussion about the surgical strategy or clinical significance. This will increase the clinical value and applicability of your research.

Response

Based on our results in this study, we have created a prognostic prediction formula using three factors and classified the calculated results into three categories as follows:

Prognostic prediction formula: 1.33 × 10-5 CA19-9 + 2.03 CAR – 0.48 CRP.

-1.9 × 10-1 to 9.2 × 10-4: Worse prognosis

9.2 × 10-4 to 1.9 × 10-2: Moderate prognosis

1.9 × 10-2 to 4.0: Better prognosis

We believe that it is very useful in terms of treatment strategy to clarify the prognosis based on the above prognostic prediction formula using a patient’s biochemical data before surgery. In other words, if the prognosis of a patient is predicted to worsen before surgery, it may be desirable to stop upfront surgery and perform chemotherapy. In addition, although it is necessary to consider in a prospective study, aggressive postoperative adjuvant chemotherapy may be warranted in the moderate prognosis group. However, this prognostic prediction formula represents an analysis of data from only four Japanese universities, and we believe that it is necessary to obtain a prognostic prediction formula using an analysis of data from more facilities.

We added these contents to the Discussion section, as follows:

Discussion (lines 271–282)

We constructed a prognostic formula based on the Cox proportional hazards model, as follows: 1.33 × 10-5 CA19-9 + 2.03 CAR – 0.48 CRP. In addition, they were classified into the following three groups based on the results of the prognostic prediction formula: -1.9 × 10-1 to 9.2 × 10-4, worse prognosis; 9.2 × 10-4 to 1.9 × 10-2, moderate prognosis; 1.9 × 10-2 to 4.0, better prognosis. If the prognosis of a patient was predicted to worsen before surgery, it may be desirable to stop upfront surgery and perform chemotherapy. In addition, although it is necessary to consider in a prospective study, aggressive postoperative adjuvant chemotherapy may be warranted in the moderate prognosis group. However, this prognostic prediction formula and the results of three numerical classifications were predictions based on the results of our surgery, which was performed in only a small number of cases, and it is necessary to evaluate this formula by examining a large number of cases in the future.

  1. Moreover, minor editing and language revisions are needed.

Response

The text, tables, and figures have been re-edited by an English proofreading company.

Reviewer 2 Report

The authors investigated preoperative serum markers and risk classification in intrahepatic cholangiocarcinoma (ICC) by a multicenter retrospective study.

This study is considered a breakthrough in estimating the prognostic risk of ICC without patient invasion.

Progress in effective chemotherapy for ICC has yet to be made, and the development of effective recurrence prevention therapy for moderate- and high-risk groups is eagerly awaited.

Therefore, I will be able to recommend it for acceptance.

Author Response

We thank the reviewer for the valuable comments.

Reviewer 3 Report

Overall, this is a clear, concise, and well-written manuscript. The authors used retrospective surgical & clinical data on preoperative blood test biomarkers to create a prognostic classification system using CART analysis. The introduction is relevant and focuses on the previous study characteristics. Sufficient information about the previous study findings is presented for readers to follow the present study rationale and procedures. The methods are generally appropriate, although the addition of a few variables would strengthen the conclusions if available. Overall, the results are clear and compelling.

The two major conclusions reported by the authors are:

1. CART analysis using results of multivariate analysis of biomarkers revealed that prognosis could be predicted with CA-19, CRP, and CAR. Five-year rates differed significantly (P<0.001) between groups: 60.4% (low-risk), 22.8% (moderate-risk), and 4.1% (high-risk) for RFS and 69.2% (low-risk), 32.3% (moderate-risk), and 9.2% (high-risk) for OS.

2. No difference in OS  was observed between patients in the low-risk group with or without postoperative adjuvant chemotherapy, although it improved in the moderate group and was prolonged significantly in the high-risk group receiving this chemotherapy.

The authors make a systematic contribution to the research literature in this area of investigation. Overall, this is a high-quality manuscript with implications for the basis, development, and implementation of high-value research in the management of ICC patients. Specific comments follow.

Major Comments:

1.       Line 66-68 – Please consider rephrasing this to indicate that analyzing these markers using CART and machine learning could improve the accuracy of the biomarkers to prognosticate ICC.

2.       Please comment on why blood loss was significant to recurrence-free survival.

3.       Is there an easy formula to use incorporating CA19-9, CAR, and CRP to stratify into 3 groups? A score calculator might be a good tool for future research.

4.       When were the blood markers drawn? Please comment on the timing of the blood draw and surgery. 

Author Response

We thank the reviewer for the valuable comments.

Major Comments:

  1. Line 66-68 – Please consider rephrasing this to indicate that analyzing these markers using CART and machine learning could improve the accuracy of the biomarkers to prognosticate ICC.

Response

According to the reviewer’s comments, we have revised the text as follows:

Introduction (lines 70–75)

Classification and regression tree (CART) analysis is a nonparametric decision tree technique that forms a collection of rules based on variables that can divide three populations into groups. CART analysis has gradually been incorporated into cancer prognosis prediction [20-22], and we have used it here to apply to a population of patients with ICC. Analyzing these indicators using CART and machine learning may improve the accuracy of biomarkers predicting ICC.

  1. Please comment on why blood loss was significant to recurrence-free survival.

Response

Please see Table 2. Blood loss greater than 540 mL was significant (p < 0.001) in univariate analysis but not significant (p = 0.24) in multivariate analysis for recurrence-free survival.

  1. Is there an easy formula to use incorporating CA19-9, CAR, and CRP to stratify into 3 groups? A score calculator might be a good tool for future research.

Response

Based on our results in this study, we have created a prognostic prediction formula using three factors and classified the calculated results into three categories as follows:

Prognostic prediction formula: 1.33 × 10-5 CA19-9 + 2.03 CAR – 0.48 CRP.

-1.9 × 10-1 to 9.2 × 10-4: Worse prognosis

9.2 × 10-4 to 1.9 × 10-2: Moderate prognosis

1.9 × 10-2 to 4.0: Better prognosis

We believe that it is very useful in terms of treatment strategy to clarify the prognosis based on the above prognostic prediction formula using a patient’s biochemical data before surgery. In other words, if the prognosis of a patient is predicted to worsen before surgery, it may be desirable to stop upfront surgery and perform chemotherapy. In addition, although it is necessary to consider in a prospective study, aggressive postoperative adjuvant chemotherapy may be warranted in the moderate prognosis group. However, this prognostic prediction formula represents an analysis of data from only four Japanese universities, and we believe that it is necessary to obtain a prognostic prediction formula using an analysis of data from more facilities.

We added these contents to the Discussion section, as follows:

Discussion (lines 271–282)

We constructed a prognostic formula based on the Cox proportional hazards model, as follows: 1.33 × 10-5 CA19-9 + 2.03 CAR – 0.48 CRP. In addition, they were classified into the following three groups based on the results of the prognostic prediction formula: -1.9 × 10-1 to 9.2 × 10-4, worse prognosis; 9.2 × 10-4 to 1.9 × 10-2, moderate prognosis; 1.9 × 10-2 to 4.0, better prognosis. If the prognosis of a patient was predicted to worsen before surgery, it may be desirable to stop upfront surgery and perform chemotherapy. In addition, although it is necessary to consider in a prospective study, aggressive postoperative adjuvant chemotherapy may be warranted in the moderate prognosis group. However, this prognostic prediction formula and the results of three numerical classifications were predictions based on the results of our surgery, which was performed in only a small number of cases, and it is necessary to evaluate this formula by examining a large number of cases in the future.

  1. When were the blood markers drawn? Please comment on the timing of the blood draw and surgery. 

Response

According to the reviewer’s comments, we have added the following text to the revised manuscript:

Materials and Methods (lines 99 –101)

At all institutions participating in this study, blood sampling for blood test parameters was performed from approximately 2 weeks to the day before surgery.

Reviewer 4 Report

Comments
This manuscript is quite interesting and good chronological, and precious study for patients. However, there are some comments for improvement below:

1. Figure 1: Should state how come do the authors cuff-off at 0.042 for CAR and 0.17 for CRP.

2. The revised AJCC 8th edition incorporated a tumor size cutoff of 5 cm, why did the authors cut -off the tumor size at 3.5 cm?

3. There are lots of redundant texts for explaining the table, please revise.

4. The author discussed that the values of CA19
-9, CRP, and CAR as risk classifications for RFS.based on machine-based approach. Should add on more in the obvious
correlation with three risk groups

Author Response

We thank the reviewer for these valuable comments.

  1. Figure 1: Should state how come do the authors cuff-off at 0.042 for CAR and 0.17 for CRP.

Response

The CART algorithm split the data based on all available variables and classified based on the Gini Index and entropy criteria [Grajski 1986; new ref. no. 32]. The CART algorithm was applied to classify patients into subgroups with similar prognoses using suitable factors classified based on cutoff values from ROC analysis. Using the ROC curve analysis and CART algorithm, patients were separated into three subgroups according to preoperative characteristics. Twenty-three preoperative prognostic factors were extracted for the analysis, yielding the decision tree for recurrence with the top three factors. The ROC curves and cutoff values for the three factors are shown in Supplementary Figure 1.

We have added the following sentences and Supplementary Figure 1 to the revised manuscript:

Results (lines 132–145)

  1. Results

3.1. Preoperative staging system using the CART algorithm and patients’ perioperative characteristics

Using preoperative prognostic factors and survival data, the CART algorithm divided patients into subgroups based on clinical prognosis. Twenty-three preoperative prognostic factors were extracted for the analysis, yielding a decision tree for recurrence with the top three factors (Figure 1). For RFS, CA19-9 was the first node, and CAR and CRP were the second nodes. The ROC curves and cutoff values for the three factors are shown in Supplementary Figure 1. The numbers of patients in the low-, moderate-, high-risk groups were 66, 98, and 61, respectively. The three risk classifications of patient characteristics are shown in Table 1. There were significant differences among the three groups in the following items: WBC, neutrophil, albumin, ALT, CRP, PNI, CAR, ALBI, CEA, CA19-9, type of hepatectomy, lymph node status, maximum tumor size, major vascular invasion, and AJCC staging.

Supplementary Figure 1

  1. The revised AJCC 8th edition incorporated a tumor size cutoff of 5 cm, why did the authors cut -off the tumor size at 3.5 cm?

Response

A tumor diameter of 3.5 cm was used as the cutoff value for ROC.

3.There are lots of redundant texts for explaining the table, please revise.

Response

The text, tables, and figures have been re-edited by an English proofreading company.

  1. The author discussed that the values of CA19-9, CRP, and CAR as risk classifications for RFS.based on machine-based approach.Should add on more in the obvious correlation with three risk groups. 

Response

Based on our results in this study, we have created a prognostic prediction formula using three factors and classified the calculated results into three categories as follows:

Prognostic prediction formula: 1.33 × 10-5 CA19-9 + 2.03 CAR – 0.48 CRP.

-1.9 × 10-1 to 9.2 × 10-4: Worse prognosis

9.2 × 10-4 to 1.9 × 10-2: Moderate prognosis

1.9 × 10-2 to 4.0: Better prognosis

We believe that it is very useful in terms of treatment strategy to clarify the prognosis based on the above prognostic prediction formula using a patient’s biochemical data before surgery. In other words, if the prognosis of a patient is predicted to worsen before surgery, it may be desirable to stop upfront surgery and perform chemotherapy. In addition, although it is necessary to consider in a prospective study, aggressive postoperative adjuvant chemotherapy may be warranted in the moderate prognosis group. However, this prognostic prediction formula represents an analysis of data from only four Japanese universities, and we believe that it is necessary to obtain a prognostic prediction formula using an analysis of data from more facilities.

We added these contents to the Discussion section, as follows:

Discussion (lines 271–282)

We constructed a prognostic formula based on the Cox proportional hazards model, as follows: 1.33 × 10-5 CA19-9 + 2.03 CAR – 0.48 CRP. In addition, they were classified into the following three groups based on the results of the prognostic prediction formula: -1.9 × 10-1 to 9.2 × 10-4, worse prognosis; 9.2 × 10-4 to 1.9 × 10-2, moderate prognosis; 1.9 × 10-2 to 4.0, better prognosis. If the prognosis of a patient was predicted to worsen before surgery, it may be desirable to stop upfront surgery and perform chemotherapy. In addition, although it is necessary to consider in a prospective study, aggressive postoperative adjuvant chemotherapy may be warranted in the moderate prognosis group. However, this prognostic prediction formula and the results of three numerical classifications were predictions based on the results of our surgery, which was performed in only a small number of cases, and it is necessary to evaluate this formula by examining a large number of cases in the future.
